# Confounding Rules Can Hinder Conservation: Disparities in Law Regulation on Domestic and International Parrot Trade within and among Neotropical Countries

**DOI:** 10.3390/ani12101244

**Published:** 2022-05-12

**Authors:** Pedro Romero-Vidal, Martina Carrete, Fernando Hiraldo, Guillermo Blanco, José L. Tella

**Affiliations:** 1Department of Physical, Chemical and Natural Systems, Universidad Pablo de Olavide, 41013 Sevilla, Spain; mcarrete@upo.es; 2Department of Conservation Biology, Doñana Biological Station CSIC, 41092 Sevilla, Spain; hiraldo@ebd.csic.es (F.H.); tella@ebd.csic.es (J.L.T.); 3Department of Evolutionary Ecology, Museo Nacional de Ciencias Naturales CSIC, 28006 Madrid, Spain; gblanco@mncn.csic.es

**Keywords:** conservation, law regulations, parrots, poaching, wildlife crime, wildlife legislations, wildlife trade

## Abstract

**Simple Summary:**

Wildlife trade represents one of the main causes of biodiversity loss worldwide. In an attempt to control this practice, both international and national legislation has been adopted to regulate trapping and trade in wild animals. For parrots, one of the most traded bird orders, the Convention on International Trade in Endangered Species of Wild Fauna and Flora (CITES) has regulated their international legal trade since 1975. However, wildlife regulations within Neotropical countries—the main sources for the past international trade—vary widely and differ from the international ones. This complex legislative framework translates into a lack of knowledge on the legal status of this activity in many countries, including within the scientific community. This confusion may be increasing the conservation problems of many vertebrate groups.

**Abstract:**

Wildlife trade is a major driver of biodiversity loss worldwide. To regulate its impact, laws and regulations have been implemented at the international and national scales. The Convention on International Trade in Endangered Species of Wild Fauna and Flora (CITES) has regulated the international legal trade since 1975. However, an important volume of illegal trade—mainly within countries—continues to threaten several vertebrate groups, which could be due to a lack of specific legislation or enforcement of existing regulations. Our aim was to gain a more accurate picture of poaching and legal possession of native parrots as pets in the Neotropics, where illegal domestic trade is currently widespread. We conducted a systematic search of the laws of each of the 50 countries and overseas territories, taking into account their year of implementation and whether the capture, possession and/or sale of parrots is permitted. We compared this information with legal exports reported by CITES to assess differences between the enforcement of international and national trade regulations. We found that only two countries (Guyana and Suriname) currently allow the capture, trade and possession of native parrots, while Peru allowed international legal trade until recently. The other countries have banned parrot trade from years to decades ago. However, the timing of implementation of international and national trade regulations varied greatly between countries, with half of them continuing to export parrots legally years or decades after banning domestic trade. The confusion created by this complex legal system may have hindered the adoption of conservation measures, allowing poaching, keeping and trade of protected species within and between neighboring countries. Most countries legally exported Neotropical parrot species which were not native to those countries, indicating that trans-border smuggling often occurred between neighboring countries prior to their legal exportations, and that this illicit activity continues for the domestic trade. Governments are urged to effectively implement current legislation that prohibits the trapping and domestic trade of native parrots, but also to develop coordinated alliances and efforts to halt illegal trade among them. Otherwise, illegal trade will continue to erode the already threatened populations of a large number of parrot species across the Neotropics.

## 1. Introduction

Wildlife trade, one of the main drivers of global defaunation [1], is a lucrative activity in which millions of individuals are captured and moved annually legally or illegally around the world [2]. The creation of the Convention on International Trade in Endangered Species of Wild Fauna and Flora (CITES) has led to a significant decrease in the number of exports of species included in its appendixes at an international level [3,4]. However, the implementation of CITES at a national scale by the different parties (each one of the signatory countries) results in a complex web of national laws and regulations [5], which often deal differently with the international and domestic trade in each country. These differences may lead to misinterpretations when applying regulations at a more local scale, resulting in illegal trade that is not perceived as such. For example, recent wildlife market surveys conducted in countries where international and domestic trade coexist have shown that both CITES regulations and harvest and trade quotas are often violated, and that disentangling legal and illegal trade is a difficult task [6]. It is of great concern that mischaracterization of legal and illegal wildlife trade and its impacts on the conservation of traded species can mislead policy processes [7].

The order Psittaciformes (parrots and allies) is the most traded group of birds for use as companion pets [8,9]. The international legal trade of wild-caught parrots has been greatly reduced and gradually replaced by captive-bred individuals, especially after the US and EU bans [5,10]. Although there continues to be significant figures of illegal parrot trade at the international level [11,12], these numbers are no longer comparable to those of legal exports, implying a great reduction in absolute terms of wild-sourced parrots traded internationally. However, in source countries, law enforcement does not appear to have such a significant effect on reducing illegal domestic parrot trade [13].

The Neotropics constitutes the second most diverse realm in terms of parrot species [14]. Since the implementation of CITES in 1975, South America was the largest exporting area of wild-sourced parrots, but the export quantity declined sharply after 1992 [5]. However, illegal domestic trade is still widespread [15,16,17] and continues to represent an important threat for parrot populations [18]. Thousands of parrots are sold annually in major wildlife city markets [19,20,21]. Moreover, these numbers may represent only a small percentage of what is annually poached when considering rural areas where pet parrots are not acquired in city markets but are locally trapped [22,23,24,25], and the high mortality during capture and transport before selling them [26]. The complicated legislative framework in these countries [5], as well as the limited resources to enforce legislation [27] may constitute determinant factors driving these high figures of illegal parrot trade in the Neotropics, where keeping parrots and other wild animals as pets is a rooted tradition [28].

A recent worldwide survey of experts on bird trade concluded that the main challenges in reducing illegal trade were its poor monitoring and assessment, and the lack of legislation enforcement and environmental awareness among stakeholders [29]. Similar problems were evidenced by a literature survey on illegal parrot trade in the Neotropics and other regions of the world [13]. Moreover, the ignorance of laws and regulations by local people as an explanation for poaching and keeping wild animals in these areas is rarely addressed. In addition to other factors such as poverty [30], limited resources for local authorities, and/or corruption [27], locals may be unaware of national wildlife laws. As an example, in a study conducted in Venezuela based on interviews, most local people had negative attitudes towards poaching parrots for selling but positive ones towards keeping them as household pets [24]. In other areas, perception of local people on the illegality of poaching and keeping parrots as pets is almost inexistent [23,25]. The lack, or incorrect knowledge, of these illegal activities may be increased by the fact that in some Amazonian countries, domestic trade was prohibited while trapping for the international parrot trade was still allowed, sometimes focusing on a few species under annually varying harvest quotas [17]. Even the perception by the scientific community of this activity may be erroneous, in some cases interpreting the domestic trade of certain species as legal [20] when in fact it does not fulfill the criteria established by national laws [17]. Moreover, differences in legislation and law enforcement between neighboring countries may have also promoted illegal parrot trade between them [17].

The complex web of legislation concerning trapping and keeping wild fauna in the Neotropics makes it necessary to clarify the legal situation in each country in order to better understand the magnitude of this harmful activity for wildlife. This paper constitutes the first attempt to document the extant wildlife law legislation for each country and territory in the Neotropics, focusing on parrots. In addition, our comparison of national and international regulations serves to verify whether there is a temporal disparity in regulation within and among countries in the Neotropical region. Our results show that only two countries currently allow the trapping, trade and keeping of wild-caught parrots. Moreover, there are significant discrepancies between the year of application of national and international trade regulations. These discrepancies increase the current confusion about the legal trade status of many species, which combined with other factors such as governments’ lack of resources to pursue appropriate legal actions, makes it even more challenging to curb illegal parrot trade, which constitutes a major conservation threat for Neotropical parrots.

## 2. Materials and Methods

### 2.1. Systematic Legislation Search

To review existing national legislation dealing with the trapping, domestic trade and keeping of native parrots as pets, a systematic search was carried out using ECOLEX (ecolex.org, accessed on 21 September 2021), an online environmental law search platform. This systematic search was conducted for all the 50 Neotropical countries and overseas territories, including Central America, the Caribbean and South America. To design the search, we used the name of the territory or country, and the terms “wildlife trade” and “poaching”. First year of prohibition of capture, trade and possession of parrots (specified for this bird group or as a general prohibition for wildlife) was identified at both international and domestic levels. We supplemented these data with information provided by local experts on parrots and illegal trade, who helped to clarify the information on dates of prohibition for some of the countries and territories.

### 2.2. International Legal Trade

From the website of CITES (cites.org, accessed on 21 September 2021), we extracted the first and last year of legal export of wild-caught parrots for each of the Neotropical countries and overseas territories. In addition, we obtained the total number of individuals exported by each country and territory between those periods. We also noted whether the Neotropical parrot species exported were native or not to the exporting country, using range distribution maps [14].

### 2.3. Statistical Analysis

The non-parametric Kruskal-Wallis test was used for testing differences among subregions (Caribbean, Central America and South America) in (1) the year in which national laws prohibited parrot trapping for the domestic demand of pets and (2) the year in which the last international export was performed, according to CITES. The relationship between both variables was tested using the non-parametric Spearman correlation, both for all countries together and for each subregion separately. Analyses were performed using SPSS Statistics v. 27 (SPSS Inc., Chicago, IL, USA). 

## 3. Results

### 3.1. Domestic Parrot Trapping and Trade

From the existing 50 Neotropical countries and overseas territories (thereafter countries) we were able to obtain information on laws regulating trapping for the domestic demand of parrot pets from all but four small territories, which are not inhabited by native parrots: Anguilla, British Virgin Islands, Netherlands Antilles, and Turks and Caicos Islands (Table 1). Among the remaining 46 countries, only two (Suriname and Guyana) currently allow this activity. The rest prohibited it in years that ranged from 1935 to 2017 (average year: 1985), with variations among subregions and some similarities between neighboring countries (Figure 1). Caribbean countries tended to halt this activity earlier than South American and Central American ones (Figure 2a), but this trend is not statistically significant (Kruskal-Wallis test, H = 3.03, *p* = 0.22). 

### 3.2. International Parrot Trade

According to CITES, 37 Neotropical countries legally exported wild-caught parrots. Currently (2022), legal international trade is only allowed in Guyana, Peru and Suriname, although their last exports took place in 2019, 2017 and 2019, respectively. Overall, last exports varied among countries between 1988 and 2019 (average year: 2003, range: 1988–2019), with apparent differences between subregions and similarities between neighboring countries (Figure 1). The last exports from South America occurred in later decades than those from Central America and the Caribbean (Kruskal-Wallis test, H = 6.73, *p* = 0.035; Figure 2b). 

According to CITES, Neotropical countries exported >4 M wild-sourced parrots from 106 species between 1975 and 2021 (Table 1). The volume of parrots exported varied in orders of magnitude between countries and subregions, being much higher in South American than in Central American and Caribbean countries (Figure 3). Exports declined drastically in the last decade and are now anecdotal (Figure 3). Notably, eight countries legally exported Neotropical parrot species not native to these countries, and 30 countries exported both native and non-native Neotropical species (Table 1). 

### 3.3. Temporal Mismatches between Legal Domestic and International Parrot Trade

Countries greatly differed in the time elapsed between prohibiting domestic trade and halting international exports (time gap), domestic trade being prohibited from 82 years before to 27 years after the last exports took place (average time gap:—18 years; 26 countries continued to export parrots 2–82 years after domestic trade prohibitions, Table 1). This time gap did not differ among subregions (Kruskal-Wallis test, H = 2.11, *p* = 0.348). Therefore, there is no overall correlation between the years in which domestic trade was banned and the cessation of exports (Spearman r = 0.01, *n* = 32, *p* = 0.956). However, this relationship varied among subregions (Figure 4), being non-significant in Caribbean (Spearman r = 0.13, *n* = 14, *p* = 0.645) and South American countries (Spearman r =−0.31, *n* = 10, *p* = 0.381) but positive and statistically significant in Central America (Spearman r = 0.81, *n* = 8, *p* = 0.015).

## 4. Discussion

Laws and international agreements to regulate international wildlife trade are essential tools for preventing wildlife over-exploitation [3,31,32]. Although some international regulations have been successful in reducing the volume of wild-sourced animals traded internationally [5], they cannot regulate domestic wildlife trade [33]. On the other hand, one of the main problems in regulating international wildlife trade is the implementation of the main regulatory mechanism (CITES), as each signatory country is a sovereign state and is responsible for implementing law enforcement measures within its territory. The complex legislative framework that is generated makes the application of these laws at a local level very complicated, as a consequence of the differences in the regulation of the laws existing in adjacent territories but belonging to different parties. This legislative scheme, coupled with the intrinsic characteristics linked to most wildlife source countries (i.e., lack of resources, corruption [27,34,35]), which may have gotten worse due to politic and socio-economic conflicts in several countries, seem to play a determining role in the high volumes of illegal wildlife trade recorded at the domestic level.

Our compilation of laws shows the complexity of the legislation to regulate the capture and trade of parrots in the Neotropics. Successive laws, numerous amendments to earlier ones and frequent ambiguity in their wording made it difficult even for us to identify the exact year in which the capture of parrots was banned in some countries, so we must recognize that some dates reflected in Table 1 may not be entirely accurate. Nonetheless, an unquestionable result is that today, only two countries fully allow the trapping and domestic and international trade of parrots, while the rest prohibited these activities from years to many decades ago. However, illegal parrot trade is still widespread in the Neotropics [13,16,17,36].

The complexity of laws regulating domestic and international trade and their disparities within and between Neotropical countries may have confused both law enforcement authorities and potential consumers, thus contributing to keeping the illegal trade alive. In this regard, a notable point of confusion is that about half of the countries maintained legal exports years or even decades after prohibiting the capture of parrots for the domestic demand. Thus, the existence of captures for legal export could have facilitated—and even camouflaged—the domestic illegal trade. Argentina promoted the Elé project for the sustainable trapping and export of blue-fronted amazons (*Amazona aestiva*) between 1998 and 2005 [37], while domestic trade was prohibited. However, this project was criticized because it could have promoted unsustainable quotas and illegal harvests [38]. Puerto Rico constitutes a particular case. The only native parrot species (*Amazona vitatta*) was fully protected since 1967, but introduced non-native parrots (up to 29 species reported in the wild, [39]) can be trapped and sold for export with a permit from local authorities. In other cases (México, Peru), the establishment of legal harvest quotas for some parrot species under—supposedly—sustainable trade criteria also promoted illegal domestic trade, as reflected by numbers of poached parrots far exceeding quotas [21,40]. Harvest quotas even confounded the monitoring of illegal domestic trade in open markets in Peru [20]. There, the permits and requirements necessary for the capture and trade of the few allowed species made them entirely destined for legal international trade, making the sale of the same species in domestic markets illegal in practice [17,21]. This exemplifies how the complex legal framework can even confound the scientific monitoring and thus underestimate the real volume of illegally traded parrots.

The fact that several parrot species make use of agriculture and are persecuted, killed and trapped as pest crops [41] complicates the scenario of illegal trade. In many cases (e.g., parakeets of genus *Psittacara* in some Andean regions), illegal domestic trade results as a by-product of trapping parrots in crops to increase the incomes of people in rural and remote areas [36]. For three decades (1968–1998), Uruguay made an exception to the general prohibition on wildlife trapping to legally allow the killing of monk parakeets (*Myiopsitta monachus*) because they were considered as crop pests. These periodically renewed exceptions, which allowed hunting but not keeping the species as a pet, might have confounded both authorities and consumers to the extent that monk parakeets are still widely live-trapped and found in the country as household pets (Authors, unpubl. data). In Argentina, similar exceptions were legislated for monk parakeets and burrowing parrots (*Cyanoliseus patagonus*), arguing crop damage, which created even more confusion as the exceptions were not applied nationally, but were legislated separately by different provinces of the country, thus varying spatially and temporally.

The difficulties in translating complex and variable legislation to local communities make it even more difficult to halt parrot poaching. In many of the source areas where parrots are trapped, either for their own consumption as pets or for subsequent sale locally or at wildlife markets, people have little or no knowledge regarding the legislative framework concerning wildlife keeping and trapping (e.g., [23,24,25]). Authorities, through awareness campaigns or direct actions mediated by the relevant environmental authorities (i.e., seizures and prosecution), are the ones that have the greatest impact in transmitting this information to the population. In some countries, NGOs also play a key role through conservation and awareness campaigns (e.g., [23,24]), although in some cases, they are more concerned with the welfare of traded animals exposed in markets than with the actual impact of poaching on the conservation of wild populations [42]. In the end, efforts seem to be insufficient, as poached parrot pets are exposed, even in large markets, to the view of authorities throughout the Neotropics. Additionally, in most cases, no action is taken [36], or seizures are biased towards certain areas or species. As an example, in Costa Rica, seizures are very targeted to certain species (mainly threatened amazons and macaws), so in many areas of the country, local people capture and keep other parrot species thinking that they are legally allowed because they are not confiscated by authorities [36]. This can pose a major conservation problem, because while conservation efforts are focused on certain species where there is a perception of a prohibition on trapping, other threatened species may be overexploited due to ignorance of their current legal status. 

The existing confusion extends not only to the species allowed or not, but also to what aspects of poaching, selling or keeping parrots are legal or illegal. For example, in Venezuela, a recent study has focused on the Yellow-shouldered Amazon (*Amazona barbadensis*), a species highly threatened by illegal trade [43], conducting inquiries to find out the perception of the local population on these aspects [24]. Thus, despite a long-term conservation program developed over the last 31 years, people identified unsustainable use as the main threat to this species, but negative perceptions were limited to sale, not harvesting or keeping as pets. In fact, another recent study in Ecuador found that local people’s concerns for parrot species focused more on outsiders coming to poach them, but not on parrot poaching per se [25]. In this case, they perceived the poaching by outsiders as a threat for the species, but not their own trapping and selling activities, or they did not even consider the fact that poaching was prohibited by law. These studies highlight the confounding perception regarding this illegal activity, which is not properly transferred from governmental authorities to the local population.

The disparity in legislation, the timing of enactment and resource allocation for law enforcement between neighboring countries may have also promoted international illegal parrot trade by smuggling poached parrots across borders to be legally exported from surrounding countries. As an example, the CITES data show that 30 blue-winged macaws (*Primolius maracana*), a species native to Brazil, were exported from Argentina in 1982 (when the species was probably already extinct in that country [44]) and 75 individuals were exported from Bolivia (where the species is absent [14]) between 1982 and 1985. Our compilation of CITES data shows that this is not an isolated case, as 38 countries legally exported non-native, wild-sourced Neotropical parrot species, indicating that illegal trade often occurred between neighboring countries prior to their legal export. It should be noted that these cases could only be detected through CITES records for those species with ranges restricted to particular countries. Therefore, the volume of transboundary parrots smuggled for the legal international trade must be much higher when considering species with large ranges spanning several countries. Cross-border smuggling even occurred between Guyana and Suriname, the only two countries that currently allow legal trade, to benefit from higher quotas and lower levies per animal exported in the later country [17]. This illegal activity occurred not only to supply the international legal trade, but also the domestic demand for pets in neighbor countries. In this sense, illegal trade has been reported from Mexico to the US [40] and from Guatemala to Mexico [45], and several parrots poached in neighboring countries were recorded in illicit markets in Bolivia and Peru [19,20]. Our empirical research on illegal parrot trade in several Neotropical countries ([36]; Authors, unpubl. data) indicates that illegal cross-border trade remains widespread. Our records involved several countries (e.g., trade from Panama to Costa Rica, from Venezuela to Colombia, from Peru to Ecuador, from Brazil to Bolivia and Uruguay, from Paraguay to Argentina), mainly affecting the most preferred and prized species such as macaws and amazons [36,46], and appears to be related to differences in law enforcement and parrot availability between countries.

The data presented above highlight the need to address the current legislative heterogeneity regarding the capture, keeping and trade of parrots (and other taxa) in the Neotropics. The current system, in which each country has different regulations, has created a complex legislative framework that contributes to trans-boundary smuggling, impedes the implementation of effective actions by local authorities and even confuses the scientific community. The already existing network of international agreements and treaties involving several countries in the Neotropics, such as Mercosur, the Amazon Cooperation Treaty and Organization of American States (the latter includes the US and Canada), could contribute to the establishment of common legislations to address the issue of wildlife trade, homogenizing wider geographical areas and preventing further trans-boundary illegal trade in the future.

## 5. Conclusions

Laws and regulations at both the international and national scale are two of the most powerful tools for preventing the pervasive impacts caused by wildlife trade. However, their implementation at the local level remains a concern due to factors such as corruption, lack of resources and lack of knowledge of the intricate legal system. Knowledge of the status of poaching, trade and wildlife possession in each country is necessary to advance the conservation of affected species, which will allow government authorities and NGOs to work more efficiently in raising awareness and transmitting this information to the local population. The implementation of regulations without solving problems such as corruption of the authorities or lack of resources is insufficient, but greater knowledge of this complex legislative network can help in the essential points to combat these illegal practices, such as raising awareness among the local population or in the scientific community’s evaluations of these impacts. Neotropical countries are urged to effectively enforce current legislations prohibiting the capture and trade of native parrots—and other wildlife—for the domestic pet demand, but also to develop partnerships and coordinated efforts to curb illegal trade among themselves. Otherwise, illegal trade will continue to erode the already threatened populations of many parrot species across the Neotropics. 

## Figures and Tables

**Figure 1 animals-12-01244-f001:**
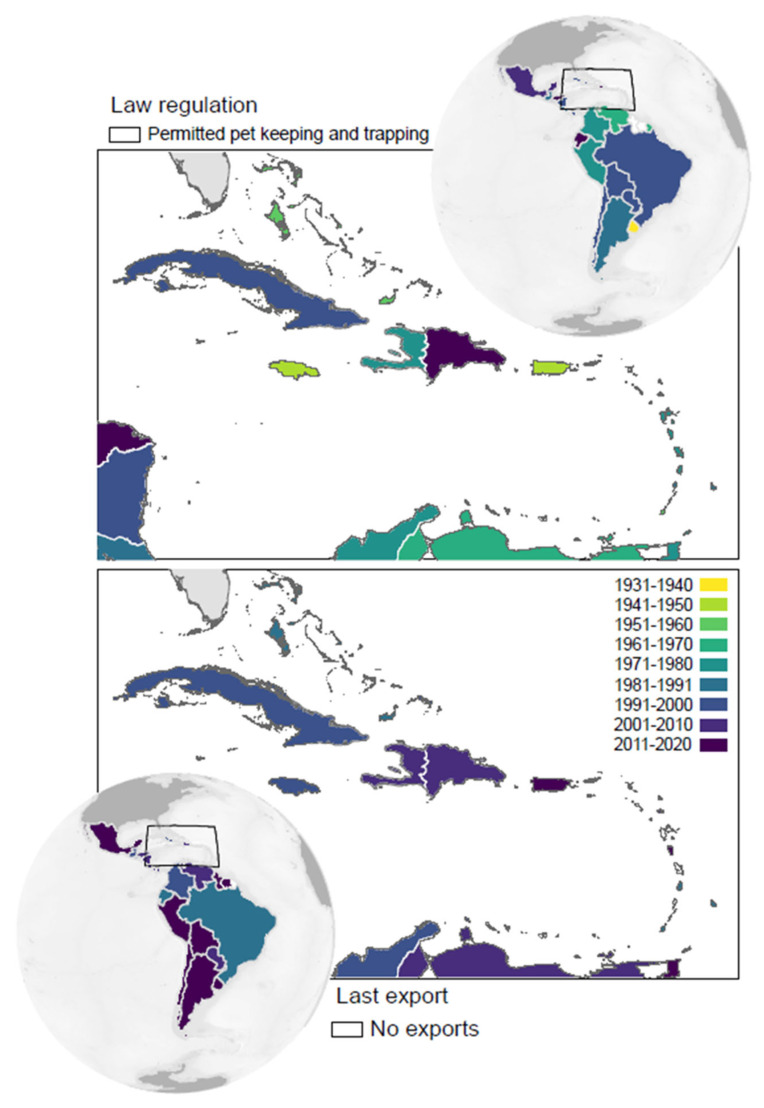
Years (grouped into decades) in which each country prohibited trapping of wild parrots for the domestic demand for pets (**top maps**) and in which decade countries performed the last export of wild-caught parrots for the international pet markets (**bottom maps**).

**Figure 2 animals-12-01244-f002:**
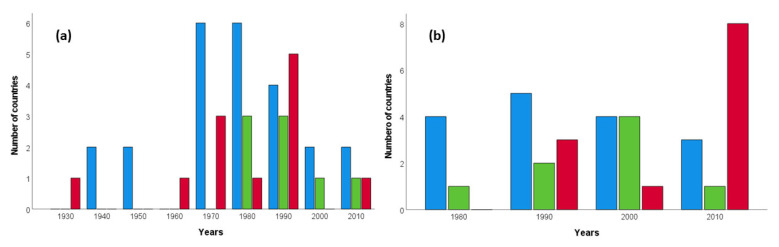
Years (grouped into decades) in which Neotropical countries (**a**) prohibited trapping wild parrots for the domestic demand of pets and (**b**) performed the last legal exports of wild-caught parrots for the international pet markets. Countries are grouped as Caribbean (blue bars), Central American (green bars) and South American (red bars).

**Figure 3 animals-12-01244-f003:**
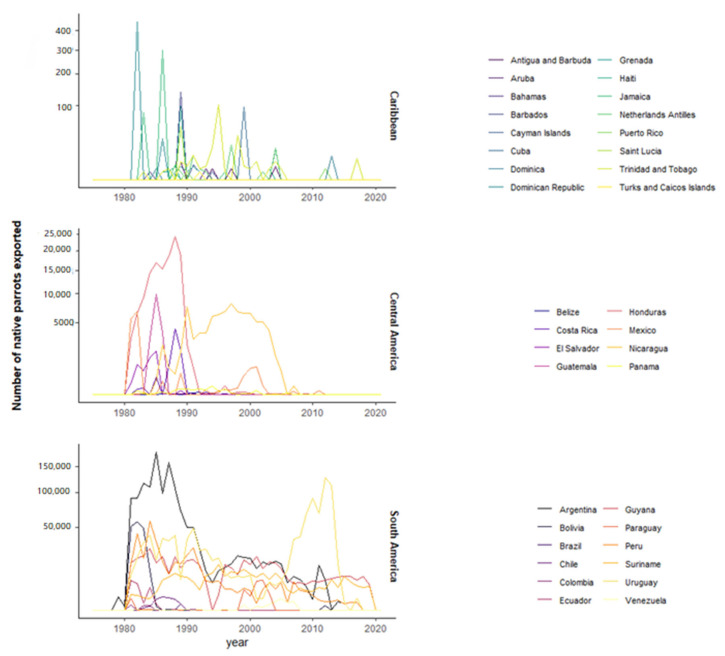
Number of wild-sourced parrots annually exported from each Neotropical country, grouped by subregions. Note different scales for each subregion.

**Figure 4 animals-12-01244-f004:**
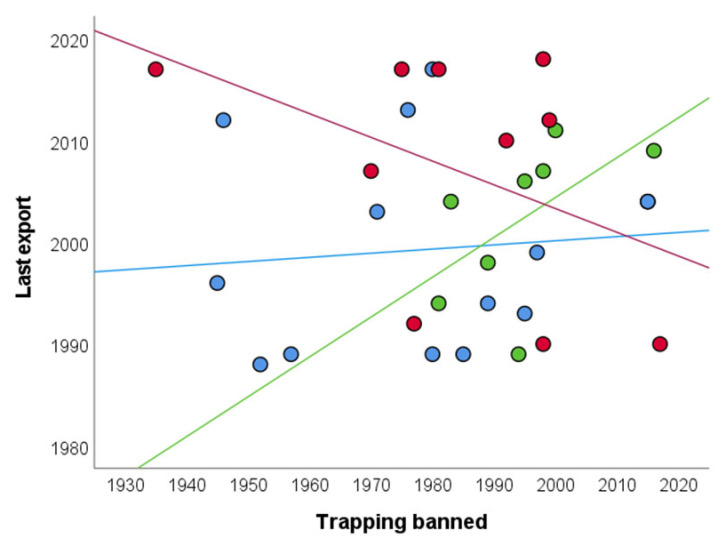
Relationships between the years in which each Neotropical country prohibited trapping wild parrots for the domestic demand of pets (*X*-axis) and the years they performed the last legal exports of wild-caught parrots for the international pet markets (*Y*-axis). Countries are grouped by subregion as Caribbean (blue), Central American (green) and South American (red). Lines represent the correlation between both variables for countries within each region.

**Table 1 animals-12-01244-t001:** Years in which each country prohibited trapping wild parrots for the domestic demand of pets (Year) and year of the last export of wild-caught parrots for the international pet markets (Last year export), number of individuals (No. of individuals) and species (No. of species) exported and the origin of these species (Origin: N = native from the country, E = Neotropical species but absent in the country and B = both cases).

Country	Native Parrots	Year	Law	Last Year Export	No. of Individuals	No. of Species	Origin
**Anguilla (UK)**	no	-	-	-	-	-	-
**Antigua and Barbuda**	no	2015	Environmental protection and management act No. 11.	2004	13	6	E
**Argentina**	yes	1981	Ley 22.421	2017	1,397,420	27	B
**Aruba**	yes	1995	Natuurbeschermingsverordening.	1993	2	1	E
**Bahamas**	yes	1952	Act Nº 52	1988	2	2	E
**Barbados**	no	1985	Wild Birds Protection Act. Ordinance No. 27.	1989	138	5	E
**Belize**	yes	1981	Wildlife Protection Act No. 4.	1994	19	6	B
**Bermuda**	no	2006	Endangered Animals and Plants Act	-	-	-	-
**Bolivia**	yes	1999	Ley 1333	2012	167,581	46	B
**Brazil**	yes	1998	LEI Nº 9.605	1990	331	8	B
**British Virgin Islands**	no	-	-	-	-	-	-
**Cayman Islands**	yes	1989	National Conservation Law	1994	6	5	E
**Chile**	yes	1998	ley 19.473	2018	4755	5	B
**Colombia**	yes	1977	Resolucion 0787	1992	574	8	B
**Costa Rica**	yes	1983	Ley Nº 6.919—Ley de conservación de la fauna silvestre; substituted by Ley 7317.	2004	7343	11	B
**Cuba**	yes	1997	Ley del Medio Ambiente, Ley 81	1999	94	7	B
**Dominica**	yes	1976	Forestry and Wildlife Act.	2013	12	6	B
**Dominican Republic**	yes	2015	Ley Sectorial sobre Biodiversidad, No. 333-15. G. O. No. 10822	2004	506	9	B
**Ecuador**	yes	2014	Codigo Integral Penal Art. 247	1990	16,226	25	B
**El Salvador**	yes	1994	Ley de conservación de vida silvestre. Decreto Legislativo D Nº: 844.	1989	4801	8	B
**Falkland Islands**	no	1999	Conservation of Wildlife and Nature Ordinance No. 10	-	-	-	-
**French Guiana**	yes	1967	Loi 5197	-	-	-	-
**Grenada**	no	1957	Birds and Other Wild Life Protection Ordinance No. 26	1989	96	4	E
**Guadeloupe**	no	1977	Loi Nº 76-629. Decret Nº 77-1295.	-	-	-	-
**Guatemala**	yes	1989	Ley de Areas Protegidas	1998	16,591	12	B
**Guyana**	yes	Allowed	Wildlife Conservation and Management Bill 2016	2019	469,940	34	B
**Haiti**	yes	1971	Decret organisant la surveillance et la Police de la chasse	2003	7	2	N
**Honduras**	yes	2016	Ley de protección y bienestar animal. Decreto 115-2015.	2009	130,376	33	B
**Jamaica**	yes	1945	Wild life Protection Act.	1996	382	3	B
**Martinique**	no	1977	Loi Nº 76-629. Decret Nº 77-1295.	-	-	-	-
**Mexico**	yes	2000	Ley General de Vida Silvestre	2011	15,071	20	B
**Montserrat**	no	1996	Forestry, Wildlife, National Parks and Protected Areas Act. Act 3.	-	-	-	-
**Netherlands Antilles**	yes	-	-	2004	60	7	B
**Nicaragua**	yes	1998	Ley General del Medio Ambiente y Los Recursos Naturales. Ley No. 217. Decreto No. 8-98.	2007	86,246	18	B
**Panama**	yes	1995	Ley de Vida Silvestre de Panama. LeyNo. 24.	2006	358	18	B
**Paraguay**	yes	1992	Ley nº 96 de Vida Silvestre	2010	19,635	12	N
**Peru**	yes	1975	Decreto Ley Nº 21147—Ley forestal y de fauna silvestre.	2017	362,881	28	B
**Puerto Rico**	yes	1946	Commonwealth regulations (EYNF)	2012	3	2	E
**Saint Kitts and Nevis**	no	1987	National Conservation and Environment Protection Act No. 5	-	-	-	-
**Saint Lucia**	yes	1980	Wildlife Protection Act No. 9	1989	50	1	N
**Saint Martin (FR)**	no	1977	Loi Nº 76-629. Decret Nº 77-1295.	-	-	-	-
**Saint Vicent and the Grenadines**	yes	1987	Wildlife Protection Act	-	-	-	-
**Saint-Barthélemy (FR)**	no	1977	Loi Nº 76-629. Decret Nº 77-1295.	-	-	-	-
**Sint Maarten (NL)**	no	2003	Nature Conservation Ordinance St. Marteen. AB2003, No. 25	-	-	-	-
**Suriname**	yes	Allowed	-	2019	243,330	28	B
**Trinidad and Tobago**	yes	1980	Conservation of Wild Life Regulations. Conservation of Wildlife Act 16	2017	214	6	B
**Turks and Caicos Islands**	no	-	-	1992	1	1	E
**United States Virgin Islands**	no	1990	Species Act of 1990	-	-	-	-
**Uruguay**	yes	1935	Ley Nº 9.481—Normas sobre protección de la fauna indígena.	2017	1,054,406	11	B
**Venezuela**	yes	1970	Gaceta Oficial 29.289	2007	3324	9	B

## Data Availability

All data analyzed in this article are provided in Table 1.

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
