# Peer review of "Confounding Rules Can Hinder Conservation: Disparities in Law Regulation on Domestic and International Parrot Trade within and among Neotropical Countries"

_animals, 2022, doi:10.3390/ani12101244_

Round 1

Reviewer 1 Report

The goal of this manuscript is worthwhile. This manuscript will serve as an important contribution to the parrot conservation literature, and if successful, could help prompt governments to develop more cohesive policies to prevent parrot and wildlife poaching. I thought the data acquisition and analyses were fine and the Figures provided excellent visual representations of the data.

I recommend changing the title slightly. You either need a comma (e.g. "Confounding rules, difficult conservation" or make it "Confounding rules=difficult conservation"

An important reference was left out; “Nest Poaching in Neotropical Parrots” (Wright et al 2001) which investigated poaching rates, looked at poaching rates in comparison to conservation programs, and compared relative to passage of a major piece of legislation. Including this article is necessary to get a complete view of the relevant literature, as well to provide readers with an overall summary of how legislation can impact trade.

Overall, several areas in the writing could be tightened to be less repetitive. However, overall I think this will be a useful addition to the literature.

My comments below are all for minor things to be changed.

20-21 Rewrite: This confusion may be increasing the conservation problems of many vertebrate groups

61-66; this section could use some clarification. You say that “it” is difficult to disentangle between the legal and illegal trade. What is “it”? Also, in what way is wildlife trade being mischaracterized? I think more elaboration here would help.

67; Remove “considering”

98-101; Rephrase this sentence. I recommend; “Other factors that may contribute to poaching include poverty, limited resources for local authorities, and/or corruption. In addition, locals may be unaware of local laws.

165, 183; replace “similitudes” with similarities

247: replace “had” with “have”

286-289; replace “null” with “no”

305-319; I recommend using this as a transition sentence and starting a new paragraph.  Also, work on rephrasing this next section. It is currently clear what your message is when you begin this section. I think you are trying to emphasize that the public does not understand that poaching and keeping native species for pets has a significant impact on parrot populations. If that is your main theme, please clarify. Also consolidate section.

361; remove first comma

Reviewer 2 Report

This a nice paper, but needs a little more attention to the English wording to make the intent clearer in sections.  The effort that has gone in to identifying relevant legislation in so many countries is to be applauded.

I offer the following comments and suggested corrections.

L14 change to read '...legislation has ..'

L18 add space after 'countries - '

L21 change 'contribute to enhance' to read 'exacerbate'

L23 Delete 'several' and change 'country' to 'national'

L26 change 'threat' to 'threaten'

L29 Add full stop after 'widespread' and new sentence 'We ...'

L34 Change 'recent years' to read 'recently'

L35 Change 'since' to 'from'

L35-38 I'm not sure what the authors were trying to say here, so I can't offer any suggestions but it doesn't make much sense as it is.

L39 Change 'forbidden' to 'protected'

L41 Change 'these' to 'those'

L43 Change to read 'legislation that prohibits ...'

L69 Delete 'experts'

L79 Change 'since' to 'after'

L83-84 Clarify if Daut et al. 2015a or 2015b

L87  The authors may wish to consider including text at the end of the sentence 'Biddle et al., 2021)' regarding the often very high rates of mortality experienced during the trapping and transport process, but prior to formal export to another country.

L87 '(Chang et al., 2021' should read '(Chan et al., 2021)'

L88 Change 'low' to 'limited' and 'legislations' to 'legislation'

L97 Change 'on' to 'of'

L98 Add 'such' after 'factors'

L99 Change 'authorities low resources,' to 'authorities' limited resources'

L100 Please clarify what is meant by 'high figures'?  Do the authors mean 'large numbers' in reference to the number of parrots exported?

L100 Change 'illicitly' to 'illegality'

L103 Add 'ones' before 'towards'

L105 Removed space in 'Romero- Vidal' and change 'wrong' to 'incorrect'

L106 Delete the comma after 'that'

L107 Delete the comma after 'countries'

L111 Change 'is within legality' to read 'as legal'

L112 Change 'legislations' to 'legislation'

L115 Change 'law regulations' to 'legislation'

L116 Add 'it' after 'make;

L118 Change 'compile' to 'document' and 'regulation' to 'legislation'

L120 Add 'in regulation' after 'disparity'

L122 Delete ', being banned throughout the rest of the Neotropics.'

L126 Delete ', the' and change 'government's ' to 'governments' '

L131 Change 'legislations' to 'legislation'

L139 Change 'completed' to 'supplemented'

L140 Delete comma after 'trade'

L146 Change 'these' to 'those'

L160 Delete comma after 'countries)

L166 Delete 'in'

L167 Delete 'decades'

L175 Add 'of' after 'trapping'

L176 Change 'of' to 'for', add 'decade' after 'which' 

Figure 3  In their current format the figures (all three) are too small to allow the reader the opportunity to see any useful detail or to read the legends.

L218 Add 'by sub-region' after 'grouped'

L223-224 References for Huxley 2000 and Schneider 2012 do not appear in the reference list.

L224-226.  This sentence is illogical as international legislation cannot override the legislation of sovereign states.

L232-235 The authors might also want to consider adding text to this part of the discussion that considers the role that dictatorships and democracies compromised by major drug trade has on fauna trade.  It is common to see nations with drug or weapons trade also active in fauna trade as the three 'currencies' are interchangeable and are often transported by the same routes.  Argentina and Columba, the two big players in the parrot trade in the 1980s and 1990s both provide examples where this has been the case.

L238 Change 'make' to 'made'

L247 Change 'had' to 'have'

L251 Change 'specially' to 'especially'

L257 Change 'is' to 'was'

L262 Change 'domestic illegal' to 'illegal domestic'

L267 Clarify if Daut et al. 2015a or 2015b?

L272  Change 'domestic illegal' to 'illegal domestic'

L276 Change 'of considering them as' to read 'they were considered'

L281 Change 'damages' to 'damage'

L282 Delete comma after 'confusion'

L285 Change 'to translate' to read 'in translating'

L288 Change 'null' to 'no'

L291 I don't understand what the authors mean by 'environmental parts'??

L303 Add 'to do so' at the beginning of the sentence.

L304 Change 'that are perceived as banned' to read 'where there is a perception of a prohibition on trapping'

L309'Briceno-Linares et al., 2011' has not been included in the reference list.

L314 Change 'people's concern' to read 'peoples' concerns'

L320 Suggest changing this sentence to read something like 'The disparity in legislation, the timing of enactment and probably in level of resourcing for law enforcement between neighbouring countries ....'

L325 Add 'already' before 'extinct' and 'this' to 'that'

L327 Change 'anecdotal' to 'isolated'

L329 Change 'exports' to 'export'

L333 Delete 'undetected' and replace 'because of their' with 'with'

L355 Change 'comprising' to 'involving'

Reviewer 3 Report

This is a very well written paper collating and summarising legislation regarding parrot keeping in the Neotropics. It is a straightforward but important contribution and I could envision a series of these papers being written for various target species (and ideally translated into the languages of the region - could this be possible as an appendix?). I feel like I am reading an advanced version of the manuscript and was really pleased to find that I have very few comments. Well done to the authors for an important work that is suitable for publication in Animals.

Introduction: very clearly written - good broad scope leading to the target of the paper

Line 67-70 - make two sentences to improve clarity

Methods: good definition of Neotropical (I know this term is now controversial but I am happy with it)

The statistical test is very simple - I am not sure it is really helpful here as there are many confounding factors that could go into a more complex model to understand the nuances of the trade. This may in fact oversimplify the work and I personally feel either more variables need to be considered in a mixed model (perhaps beyond the scope of this paper), or descriptives used to describe trends rather than the chosen tests...if keeping the chosen tests best not to over interpret.

Discussion 248-269 - these are really interesting examples but a lot to unpack. Could include even more information and add three sentences per example, but certainly make each long sentence less complex and two sentences per example

Line 270 - cultivations is an odd term here - crops?

Line 302 - is there any reason why they think it is legally allowed? Could this be due to seeing parrots in shops (bred ones) or in media? Line 316 - do you mean that they consider outsiders poaching as a threat but not their own - their own being if they themselves catch? this is all very interesting but just needs clarity
